# IQ-NET: A Deep Learning Approach for Fast and Accurate Phylogenetic Inference

Chen Yang[1], Zixin Zhuang[1], Piyumal Demotte[1], Cuong Cao Dang[2], Le Sy Vinh[2], Bui Quang Minh[1], Nhan Ly-Trong[1]

[1] School of Computing, Australian National University, Canberra, 2600, ACT, Australia
[2] Faculty of Information Technology, University of Engineering and Technology, Vietnam National University, Hanoi, 144 Xuan Thuy, Cau Giay, 10000 Hanoi, Vietnam

**Abstract.** Phylogenetic inference reconstructs evolutionary relationships between species from molecular sequences. Although likelihood-based methods are rigorous, they are computationally expensive. Existing machine learning based phylogenetic approaches are faster but often trained on simulated data, which are far removed from real data, causing out-of-distribution predictions. Here, we introduce IQ-NET (Intelligent Quartet NETwork), a permutation-invariant deep learning framework trained directly on empirical alignments to estimate phylogenetic quartet trees. IQ-NET jointly infers the tree topology and branch lengths without model assumptions, achieving superior accuracy and a 24-fold speedup over the widely used IQ-TREE software.

**Keywords:** Phylogenetic inference · Machine learning · Quartet analysis · Empirical data training.

## 1 Introduction

Phylogenetic inference reconstructs evolutionary trees from multiple sequence alignments (MSAs), providing insights into species origins. Widely used maximum likelihood and Bayesian methods provide statistically rigorous inference but are time-consuming. Recently proposed machine learning models enable faster inference but depend on simulated data with poor generalization.

Quartet trees represent the fundamental building blocks of phylogenetic reconstruction. They can serve as input trees for supertree assembly [1] or as the basis for species-tree inference under gene-tree discordance [8]. In this study, we focus on quartet-level inference and introduce IQ-NET, a deep learning framework that reconstructs four-taxon phylogenies directly from gapped MSAs. Trained on empirical data, IQ-NET jointly infers topology and branch lengths without model assumptions, is permutation-invariant, and achieves superior accuracy with substantial speedups compared to IQ-TREE.

## 2    Methods

### 2.1    Training Data

The empirical data were extracted from the EvoNAPS database  [2], which comprises real alignments and corresponding phylogenetic trees inferred with IQ-TREE (v2.2.0.5). The training database comprises DNA alignments collected from several established repositories, including Lanfear's BenchmarkAlignments, PANDIT, OrthoMaM, and TreeBASE. The EvoNAPS collection encompasses both DNA and protein alignments and spans a diverse range of species, from microbes and fungi to plants, birds, turtles, and mammals. To ensure strict independence between training and evaluation, all TreeBASE-derived data were excluded from EvoNAPS during the development of IQ-NET. For the final test set, we subsequently downloaded all TreeBASE studies directly from the Tree-BASE website. Moreover, EvoNAPS contains two releases (v10c and v12a) of the OrthoMaM database, so all 14,509 alignments belonging to v10c were removed to prevent redundancy with v12a. The remaining samples from the EvoNAPS database were divided into independent training, validation, and testing sets with a ratio of 80:10:10. In addition, extracted quartet trees with branches longer than 9 substitutions per site or alignments shorter than 200 sites were removed to ensure data quality.

### 2.2    Model Design

Sequences in a quartet alignment can appear in any of 24 possible orders, which may alter site pattern frequencies and affect predictions. To ensure consistent inference, we adopted the symmetry-preserving architecture proposed by Solis-Lemus' team  [3]. The designed network architecture is shown in Fig. 1

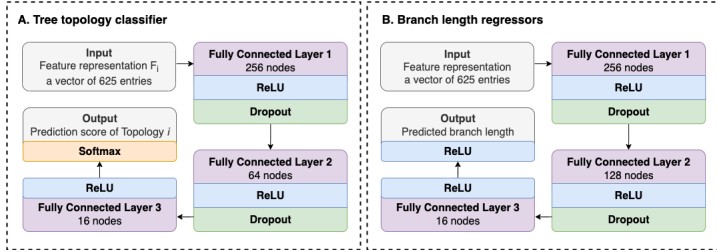

**Fig. 1.** The IQ-NET topology classifier (A) inputs a 625-dimensional site pattern frequency and outputs a softmax score for each topology. The branch-length regressor (B) has the same structure but applies ReLU activation to output the branch length.

### 2.3   Model Training

All networks were trained with the Adam optimizer and decaying learning rates. Cross-entropy and MSE losses were used for topology and branch-length prediction, respectively. Early stopping and Optuna hyperparameter tuning [4] were applied to prevent overfitting.

### 2.4   Evaluation

We evaluated IQ-NET with three key criteria. First, we benchmarked it against IQ-TREE-quartet (IQ-TREE inferring trees directly from quartet alignments) and existing machine learning methods, which include DeepNNPhylogeny [5], Fusang [6], and Suvorov-topology [7], using the testing set from the EvoNAPS database. Second, we tested IQ-NET on an independent TreeBASE dataset to assess its generalization. Finally, we demonstrate a practical application of IQ-NET by providing quartet trees to ASTRAL [8] for species tree reconstruction using the Turtle dataset [9].

## 3   Results

### 3.1   IQ-NET Outstanding Performance on EvoNAPS Test Data

IQ-NET topology classifier achieved an accuracy of 82.3%, followed by IQ-TREE-quartet (79.9%), DeepNNPhylogeny (79.4%), Fusang (76.6%), and Suvorov-topology (57.7%). We further assessed how internal branch length and sequence length influence prediction accuracy. As expected, increasing either parameter enhances accuracy across all methods by providing stronger evolutionary signal. Branch lengths predicted by the IQ-NET regressor had a Pearson correlation of 0.9007 and a slope of 0.8112, compared with 0.6754 and 0.6905 for IQ-TREE-quartet (Fig. 2). The average branch score distance (BSD; the lower the better) of IQ-NET was 0.0946, compared with 0.1485 of IQ-TREE-quartet.

Regarding runtime, all tests were conducted on the Gadi supercomputing system at the Australian National Computational Infrastructure (NCI), using a single Intel i5-14600 CPU with 4 GB of allocated RAM. IQ-TREE-quartet took 2.67 hours to reconstruct 51,241 quartet trees, whereas IQ-NET required only 6.7 minutes—a 24-fold speedup. For fairness, it should be noted that IQ-NET required initial training, which took 13.75 and 15.37 GPU minutes on Gadi for the tree topology classifier and branch length regressor, respectively. In addition, to determine the best combination of hyperparameters, we tuned the models with Optuna for 200 trials, which took 49.15 hours and 11.44 hours for the topology classification model and the branch length prediction model, respectively.

### 3.2   IQ-NET Generalizes Well on TreeBASE

We sampled 10% of quartet trees from TreeBASE [10] to benchmark topology prediction. IQ-NET achieved 82.8% accuracy versus 83.7% for IQ-TREE-quartet, demonstrating strong generalization to independent empirical data.

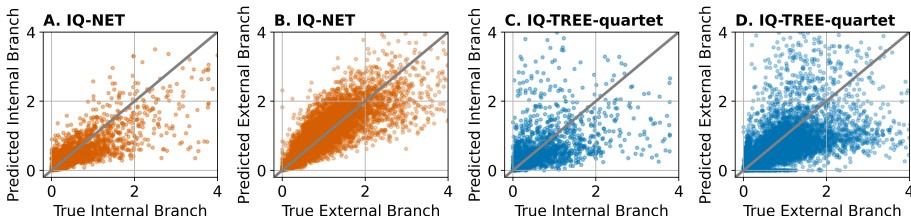

**Fig. 2.** True vs. Predicted branch lengths for IQ-NET (A, B) and IQ-TREE (C, D).

### 3.3   IQ-NET with ASTRAL Recovers Turtle Phylogeny

We used the Turtle dataset [9] to demonstrate IQ-NET combined with ASTRAL for species tree reconstruction. Following Chiari [9], we removed third codon positions, sub-sampled quartet alignments, and used IQ-NET to infer quartet trees as ASTRAL input. This approach correctly placed turtles as the sister clade to birds and crocodiles, consistent with the original study (Fig. 3).

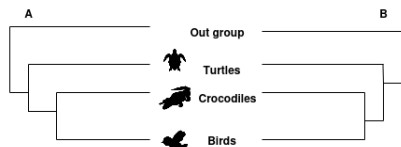

**Fig. 3.** (A) Accepted tree for placement of turtles as sister clade to birds and crocodiles [9]. (B) Tree constructed under methods of IQ-NET+ASTRAL,ASTRAL using gene trees estimated with IQ-TREE and MixtureFinder.

## 4   Conclusion

We present IQ-NET, an end-to-end machine learning framework for quartet tree reconstruction trained explicitly on real alignments. IQ-NET jointly infers the tree topology and branch lengths without model assumptions. IQ-NET surpasses existing machine learning methods and the baseline IQ-TREE in both accuracy and efficiency with a 24-fold speedup. As IQ-NET is currently trained exclusively on the EvoNAPS dataset, incorporating larger and more taxonomically diverse empirical datasets may further enhance its generalizability and robustness.

Extending IQ-NET directly to larger taxon sets remains challenging due to the factorial growth of tree topologies. A more promising direction is to integrate IQ-NET with quartet-based approaches [1], which are well suited for scaling reconstruction to larger trees.

## Acknowledgements

This work was financially supported by Chan-Zuckerberg Initiative grants for essential open-source software for science (EOSS4-0000000312 to B.Q.M.). Computational resources were provided by the Australian Government through the National Computational Infrastructure (NCI) under the ANU Merit Allocation Scheme.

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
