# OpenReview forum: "IQ-NET: A Deep Learning Approach for Fast and Accurate Phylogenetic Inference"
_AJCAI/2025/Workshop/AIML-CEB — AIML-CEB 2025 Oral_

### Official Review · Reviewer_KX24 · 2025-11-07

**Rating:** 9
**Confidence:** 4

**Review:**

This paper introduces IQ-NET, a permutation-invariant deep learning model for phylogenetic inference that reconstructs quartet trees directly from empirical multiple sequence alignments (MSAs). IQ-NET has high practical significance for molecular evolution and bioinformatics, offering a scalable, accurate alternative to computationally expensive likelihood-based inference.

It opens a path for deep learning models trained on real-world phylogenomic data, bridging the gap between ML and traditional evolutionary biology workflows. The system could inspire further applications in large-scale species tree reconstruction or integration into tools like ASTRAL.

IQ-NET represents a meaningful and timely advancement in ML–based phylogenetic inference. The combination of speed, accuracy, and empirical grounding makes it a compelling contribution for the workshop.

**Suggestions for Improvements**

1. **Computational Resources.** The 24x speedup is impressive, but including hardware specifications and training time would help readers assess the computational efficiency more rigorously.
2. **Dataset Diversity.** The model is trained primarily on EvoNAPS, which may not capture the full diversity of evolutionary signals found across different organisms.
3. **Interpretability of Learned Features.** The paper demonstrates strong performance but offers limited insight into what the network learns, for example, whether it captures evolutionary constraints, substitution patterns, or alignment length effects.

---

### Official Review · Reviewer_1Gr7 · 2025-11-08
**IQ-NET is a deep learning model for phylogenetic relationship prediction from multiple sequence alignment, outperforms leading software in performance and speed**

**Rating:** 7
**Confidence:** 2

**Review:**

The authors describe a deep learning model to predict phylogenetic relationships from multiple sequence alignments (MSAs) of DNA. Trained on data generated by competing software IQ-TREE and public data. They exclude TreeBASE and OrthoMaM as independent hold out test sets. They use an existing architecture described by Solis-Lemus (symmetry preserving).

Evaluation: 1) They compare their method to a top existing method IQ-TREE-quartet, and also 3 deep learning methods. 2) They also test their method on the independent TreeBASE dataset (assess generalisation). 3) Finally they provide a practical example on a Turtle dataset. Their method seems to show better performance, but importantly the speed is drastically improved compared to the competitor tool (~2.67 hours vs. ~6.7 mins). The generalisation is better than IQ-TREE on TreeBASE. Practical example also provided the expected results.

Comments:
* Results seem impressive in terms of performance but also particularly the speed increase
* It’s unclear how the train/test data was split: or is Fig2 showing results on the holdout set TreeBase? If this is the case, it should be clearer in the figure legend.
* Some domain specific knowledge was needed to understand the paper, for example explaining what a quartet tree is would be helpful.
* Excluding TreeBASE and OrthoMaM is helpful as a generalisation exercise, but I would still like to verify any data leakage between training data and these holdout sets. It might be domain specific knowledge that there’s no leakage, in which case I’d like a reference that shows this
* Wasn’t OrthoMaM used as a holdout set for generalisation? What was the performance of IQ-NET on this holdout set?
* I believe there’s a typo in the conclusion: it says “IQ-TREE” in the second sentence, should this be IQ-NET instead?

---

### Official Review · Reviewer_mGYU · 2025-11-11
**well executed project on predicting four taxa trees from sequence alignments with deep learning**

**Rating:** 7
**Confidence:** 4

**Review:**

## Summary
The manuscript proposes a deep learning model (IQ-NET) that constructs phylogenetic trees with four taxa
from gapped multiple sequence alignments. The output of IQ-NET provides topology and branch lengths,
and is a lot faster than IQ-TREE (a widely used model). Empirical comparisons with exisiting methods
show IQ-NET's promise.

## Evaluation
The problem is well specified, and the authors chose suitable experimental designs to check for accuracy, speedup,
and generalisation. IQ-NET is 24 times faster than IQ-TREE, and provides slightly better branch score distance.

Suggestions for improvement:
- It would be useful to also record the total time needed for training,
  (is it more than the 2.67 hours of IQ-TREE evaluation?)
  and an estimate of the effort needed for model selection.
- A discussion about the difficulty and benefits when considering more taxa (5? 6?) would be valuable.

---

### Decision · Program_Chairs · 2025-11-12

Accept (Oral)